**Investigation**

# Chromosome-specific differences in the recombination landscape of spontaneous meiotic nondisjunction

Carolyn A. Turcotte,[1] Jeff Sekelsky 🔾 [2,3,]*

[1]Curriculum in Genetics and Molecular Biology, University of North Carolina at Chapel Hill, Chapel Hill, NC 27599, United States
[2]Integrative Program for Biological and Genome Sciences, University of North Carolina at Chapel Hill, Chapel Hill, NC 27599, United States
[3]Department of Biology, University of North Carolina at Chapel Hill, Chapel Hill, NC 27599, United States

*Corresponding author: Email: sekelsky@unc.edu

Failures in chromosome segregation during meiosis result in aneuploid gametes and are the leading cause of miscarriage. The position and number of crossovers, genetic exchanges between homologous chromosomes, are essential to their accurate disjunction in meiosis. Previous research analyzing nondisjunction of acro- and telocentric chromosomes in human and *Drosophila* has identified altered positioning and number of crossovers that differs between meiosis I and meiosis II nondisjunction. Limited data from positioning in chromosomes that segregated normally in metacentric trisomies has suggested discrepancies between the behavior of these chromosomes and the acro/telocentrics in nondisjunction, which warrants further investigation. Here, we employ whole-genome sequencing to study spontaneous meiotic nondisjunction of the metacentric chromosome *2* in *Drosophila melanogaster*. Several patterns of recombination may differentially impact telo- versus metacentric chromosomes: lack of a crossover, distal crossovers, and proximal crossovers. We find that unlike meiotic nondisjunction of the *Drosophila X* chromosome, nondisjunction of chromosome *2* is not associated with dramatic changes in crossover landscape, but is associated with reduced recombination. Differences in the proportions of NDJ events with altered recombination patterns between chromosomes *X* and *2* suggest that abnormal crossover positions disparately affect chromosomes with different shapes. Taken together, these findings highlight that the underlying triggers of meiotic nondisjunction are chromosome-specific.

Keywords: meiosis, nondisjunction, recombination; crossover patterning

## Introduction

Faithful meiotic chromosome segregation ensures viable offspring in sexually reproducing species. Nondisjunction (NDJ), or failure to correctly segregate chromosomes, results in aneuploid gametes, which contribute to miscarriage. To this end, chiasmata, or physical linkages between homologous chromosomes formed during meiosis I, facilitate accurate disjunction by stabilizing the mono-orientation of homolog pairs on the meiotic spindle. Despite having this mechanism to establish correct ploidy, many organisms, including humans, have a remarkably high incidence of gametic aneuploidy (Hassold et al. 2021). A primary contributor to NDJ is incorrect regulation of crossovers, the genetic exchanges that serve as precursors to chiasmata (reviewed in Koehler et al. 1996b).

The number and position of crossovers ("crossover patterning") is tightly regulated to safeguard against NDJ (reviewed in Pazhayam et al. 2021). Major facets of crossover patterning include 1) crossover assurance, the requirement for at least one crossover per homolog pair (Darlington and Dark 1932); 2) crossover interference, a phenomenon in which crossovers do not occur in close to each other (Sturtevant 1913, 1915); and 3) pericentric suppression of crossovers (Mather 1939). Despite these phenomena first being observed over a century ago, the mechanisms behind crossover patterning and how it ensures that chromosomes correctly segregate remain unsolved.

Given the role of crossovers in correct homolog segregation, the link between NDJ and loss of crossover assurance is self-evident. The connection between crossover interference and NDJ is less clear. In theory, adjacent chiasmata should only fail to ensure correct disjunction of homologs if both crossovers occur between the same chromatids, which would limit the cohesion shared between homologs to the stretch that occurs between the two chiasmata. Pericentric crossovers may contribute to NDJ via either of two mechanisms, depending on how sister chromatid cohesion is handled in the context of a chiasma proximal to the centromere (Koehler et al. 1996a; Lamb et al. 1996; Orr-Weaver 1996). Separation of homologs requires that sister chromatid cohesion is cleaved distally to the chiasma. However, cleavage of pericentric cohesion is normally blocked until anaphase II to prevent sister chromatids from erroneously separating before the second meiotic division. If centromeric cohesion is cleaved to accomodate the proximal chiasma, pericentric crossovers will result in premature separation of sister chromatids. Conversely, if cohesion fails to be cleaved at the chiasma, homologs could become "entangled" and fail to disjoin. Recent work supported the latter model in the fission yeast *Schizosaccharomyces pombe* (Sen et al. 2024). In both cases, a meiosis I (MI) error leads to sister chromatids being

retained in the same daughter cell after meiosis II (MII), resulting in what appears to be MII NDJ.

Changes in crossover distribution in the context of meiotic NDJ have been explored in both model organisms and human trisomies. Previous work in the fruit fly *Drosophila melanogaster* revealed that crossover patterning is disparately altered in MI and MII X chromosome NDJ (Koehler et al. 1996a). Compared to normal meioses, crossovers are greatly reduced in MI NDJ meioses. The number of distal recombination events did not greatly differ between normal meioses and MI NDJ meioses, suggesting that distal chiasmata may not be as efficient in promoting disjunction. In the case of MII NDJ, crossovers are elevated exclusively near the centromere, and their numbers remain similar to those of normal meioses elsewhere. Furthermore, MI errors vastly outnumber apparent MII X chromosome segregation errors, with less than 10% of NDJ appearing to occur in MII. Very similar results to those in *Drosophila* were also found when mapping recombination events in human trisomy 21 cases of meiotic origin (Lamb et al. 1996, 1997). Notably, the *Drosophila* X and human 21 are acro-/telocentric. Data from trisomies of metacentric and sub-metacentric chromosomes in humans suggests that NDJ crossover patterning on these chromosomes may differ from that of acrocentrics (Hassold et al. 1995; Bugge et al. 1998), but the number of metacentric chromosomes and number of cases analyzed in this manner is limited, with studies of recombination in only 58 cases of meiotic NDJ for trisomy of the metacentric chromosome 16 that did not have other chromosomal abnormalities (Hassold et al. 1991, 1995), and 191 cases of meiotic NDJ for trisomy of the sub-metacentric chromosome 18 (Fisher et al. 1995; Bugge et al. 1998; Chen et al. 2005). Additionally, all of the aforementioned studies were performed via low-resolution mapping on the megabase scale.

The advent of affordable whole-genome sequencing has made it possible to map crossovers at the sub-kilobase level, with resolution limited only by polymorphism density and sequencing coverage. Crossover distribution in *Drosophila* meiosis has been explored in this manner, revealing a plethora of new information about crossover patterning, including that there are fine-scale variations in crossover rate along chromosomes, that crossover interference acts at different strengths along different chromosomes, and that double crossovers on the same chromosome arm can occur with more intimate spacing than previously appreciated (Comeron et al. 2012; Miller et al. 2012, 2016). The insights from prior studies of crossover distribution in NDJ have been numerous, but as recombination was measured via a small number of phenotypic markers (Koehler et al. 1996a) or restriction fragment length polymorphisms (Hassold et al. 1995; Lamb et al. 1996; Bugge et al. 1998), crossovers could only be mapped on the megabase scale. Greater insights about NDJ can be revealed from finer resolution of whole-genome sequencing.

Here, we analyze recombination in maternal NDJ of the metacentric chromosome 2 in *Drosophila*. We find that unlike X chromosome NDJ, chromosome 2 NDJ is associated with reduced recombination but not associated with dramatic changes in recombination landscape across the nondisjoined chromosomes. While MII NDJ was exceedingly rare for X chromosome NDJ, it comprises more chromosome 2 errors. Furthermore, MI and MII NDJ of chromosome 2 are similar in their crossover patterning. We provide evidence that these discrepancies can be attributed to differences in chromosome shape. Taken together, these results highlight that the factors contributing to meiotic NDJ are non-uniform and chromosome shape-specific.

## Materials and methods

### *Drosophila* stocks
Flies were maintained at 25°C on standard medium. Stocks obtained from the Bloomington Drosophila Stock Center (NIH P40OD018537) were used in this study. The fly stocks used for this study can be found in Supplementary Table S1 in File S1.

### Whole genome sequencing
DNA was extracted using the Monarch Spin gDNA Extraction Kit (New England Biolabs) and modified protocol for insect gDNA preparation. 20 μL 0.5 M EDTA was added to 180 μL Tissue Lysis Buffer (TLB) to make modified TLB. Using a micropestle, 20 mg of frozen whole adult flies were thoroughly crushed. 10 μL of Proteinase K (New England Biolabs) and 200 μL modified TLB were added to each crushed sample, which were mixed by vortexing. The remainder of the extraction was followed according to the standard kit protocol for extracting gDNA from animal tissue, and gDNA was eluted from the column in 100 μL gDNA Elution Buffer (New England Biolabs).

Genomic DNA from the parental stocks $w^{1118}$, *Oregon R*, and *C(2)EN, b pr* was sent to Novogene for library preparation and whole-genome sequencing and was sequenced on an Illumina NovaSeq 6000. For the NDJ progeny, libraries were prepared for sequencing using Native Barcoding Kit 24 V14 (Oxford Nanopore Technologies SQK-NBD114.24). NDJ event samples were sequenced on the Oxford Nanopore MinION Mk1B or the Oxford Nanopore GridION, as indicated in Supplementary Table S2 in File S1.

### Genomic analysis
Bowtie 2 v2.5.4 was used to align reads from the parental stocks to the dm6 reference genome. For the NDJ events, raw POD5 sequencing data was basecalled, aligned to the dm6 reference genome and demultiplexed using Dorado v1.0.0 on an NVIDIA A100 Tensor Core GPU using the basecalling model dna_r10.4.1_e8.2_400bps_sup@v5.2.0.

For all samples, a multi-way pileup was generated and variants were called using bcftools with the -m flag. The variants were filtered to isolate SNPs that were biallelic between the two parental genotypes.

### Crossover mapping
Only SNPs, and not indels, were considered for crossover mapping. The SNPs were chosen for analysis as follows: for chromosome 2, all SNPs between $w^{1118}$ and *Oregon R* were chosen as tentative candidates. From this group, any SNPs that were not homozygous in the two parental stocks were eliminated from analysis. Any SNPs that had a variance of less than 0.01 in the population of progeny were also eliminated. The same method was used for SNPs on chromosomes X and 3, but in addition any SNPs occurring in the *C(2)EN, b pr* stock were disregarded. Crossovers were identified via changes in allele frequency in the chosen SNPs along each chromosome. Of the 77 fertile NDJ males, 14 were excluded from analysis due to insufficient coverage to confidently identify crossovers on chromosome 2.

Centimorgan estimates for double crossovers were obtained by retrieving the genetic map locations of the nearest genes to each crossover via FlyBase (release FB2025_05, Öztürk-Çolak et al. 2024).

### Statistical analysis
All statistical analyses were performed in R. Two-proportion Z-tests were performed using the prop.test function and

Mann-Whitney tests were performed using the wilcox.test function.

## Results

We sought to determine what types of recombination events are more susceptible to meiotic NDJ of chromosome 2. As trisomy 2 is lethal in *Drosophila*, we resorted to a genetic trick enabling us to collect viable progeny resulting from chromosome 2 NDJ events (Fig. 1a). Females with a high polymorphism density were generated by crossing two isogenic stocks, $w^{1118}$ and *Oregon R*. These stocks have been previously used in measures of meiotic recombination (Miller et al. 2012, 2016; Crown et al. 2018). Progeny from NDJ events were obtained by mating these females to males with an entire compound chromosome 2, *C(2)EN*, in which a single centromere connects two copies of chromosome 2 in a reverse metacentric orientation (Novitski et al. 1981). In this cross, normal female meioses result in progeny that are either trisomic or monosomic for chromosome 2 and thus inviable. Viable progeny from NDJ events of chromosome 2 are either maternal 2 disomies (diplo-2 oocyte, nullo-2 sperm) or paternal 2 disomies (nullo-2 oocyte, compound-2 sperm) (Fig. 1a). These offspring are distinguishable via visible phenotypic markers on the compound 2 chromosome.

To obtain adequate material for Oxford Nanopore whole-genome sequencing, we backcrossed maternal disomy 2 males (henceforth referred to as "NDJ progeny/males") to $w^{1118}$ females and sequenced their pooled offspring. Meiotic recombination does not occur during spermatogenesis in *Drosophila* (Morgan 1910), therefore all recombination events detected in this experiment originated in the oocyte that experienced NDJ. Through this approach, we were able to analyze NDJ in 63 pairs of NDJ chromosomes (Supplementary Table S2 in File S1).

We used 294,498 single nucleotide polymorphisms (SNPs) that differed between the $w^{1118}$ and *Oregon R* genomes to identify the erroneous division (MI or MII) and crossover locations. For each NDJ male, we excluded any SNPs from analysis for which there was a sequencing depth of less than eight or a variance in the population of less than 0.01, and also excluded all SNPs on the X chromosome and chromosome 3 that were shared between the two parental genomes and the *C(2)EN* stock. This left an average of 103,515 SNPs per individual.

### Meiosis II errors make up a greater proportion of chromosome 2 meiotic NDJ than X chromosome NDJ

We first asked whether each NDJ event originated from a MI or MII segregation error. To this end, we visually analyzed allele frequencies of pericentric SNPs on chromosome 2 in each of the progeny. MI NDJ events were identified by *Oregon R*/$w^{1118}$ heterozygosity across the centromere, and conversely, Events classified as MII NDJ were identified by homozygosity for either parental genotype across the centromere (Supplementary Fig. S1 in File S1).

MI NDJ was more frequent than MII NDJ for both the X chromosome and chromosome 2 (Fig. 1b). This result is expected, as there are a wide variety of errors promoting missegregation that are exclusive to MI, such as failures in homolog pairing, synapsis, or recombination. Notably, despite MI events still comprising the majority of chromosome 2 NDJ, a significantly higher fraction of chromosome 2 NDJ events were MII in nature compared to X chromosome NDJ events (Fig. 1b, $P = 0.0018$, one-tailed two-proportion Z-test). These data indicate that the relative frequency of MI and MII NDJ are chromosome-specific.

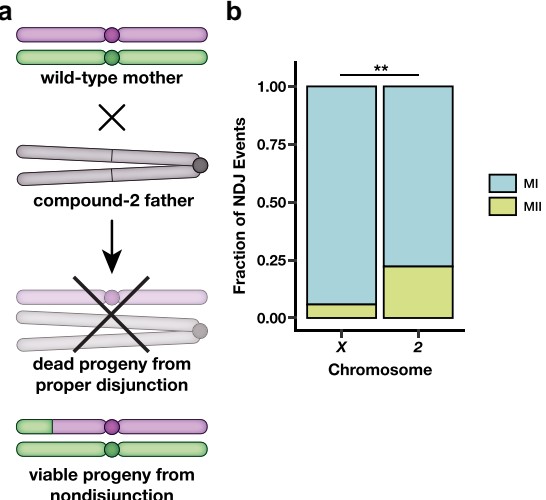

**Fig. 1.** MII errors make up a greater fraction of chromosome 2 NDJ compared to X chromosome NDJ. a) Genetic cross schematic for selection of progeny derived from chromosome 2 female meiotic NDJ events. b) The fraction of meiotic NDJ events for the X chromosome and chromosome 2 that were meiosis II (blue) or meiosis II (green) in nature. X chromosome data was taken from Koehler et al. (1996a). ** denotes $P < 0.01$ as determined by a one-tailed two-proportion Z-test. For chromosome 2 $n = 63$; for chromosome X $n = 103$.

It is worth noting that our experimental design could additionally select for progeny resulting from monosomy rescue, in which an early embryonic mitotic nondisjunction event generates a viable diploid animal from an aneuploid zygote. If this occurred, the resulting progeny would be fully homozygous across the entirety of chromosome 2 (Supplementary Fig. S1 in File S1). This outcome is only distinguishable from MII nondisjunction if progeny inherited a recombinant chromosome, in which case their genotype would switch from homozygous $w^{1118}$ to homozygous *Oregon R* (or vice-versa) at the site of recombination. Given that no such instances occur in our dataset (Supplementary Table S3 in File S1), and we have no reason to assume that a non-recombinant chromosome has a greater likelihood of rescue than a recombinant chromosome, we believe that all progeny obtained in our analysis are the result of meiotic nondisjunction events.

### Crossovers are distributed normally on chromosome 2 in both MI and MII NDJ

In *Drosophila*, X chromosome MI NDJ is associated with greatly reduced crossovers except distally, whereas MII NDJ is associated with a drastic increase in pericentric crossovers (Koehler et al. 1996a). We therefore sought to determine if these trends were consistent for chromosome 2 NDJ. To analyze the location of recombination events along chromosome 2 in the NDJ progeny, we measured allele frequency of SNPs along chromosome 2 and identified crossover locations at points along the chromosome where there was a persistent change in allele frequency (Fig. 2).

To our surprise, nearly all recombination on chromosome 2 occurred in the medial portion of the chromosome arms in both MI and MII NDJ, which is not a deviation from normal crossover patterning. Of the non-medial recombination, two crossovers from MI NDJ males were somewhat centromere-proximal, with one single crossover embedded within the pericentric heterochromatin and the other occurring at the boundary between heterochromatin and euchromatin. One crossover in a MII NDJ male also

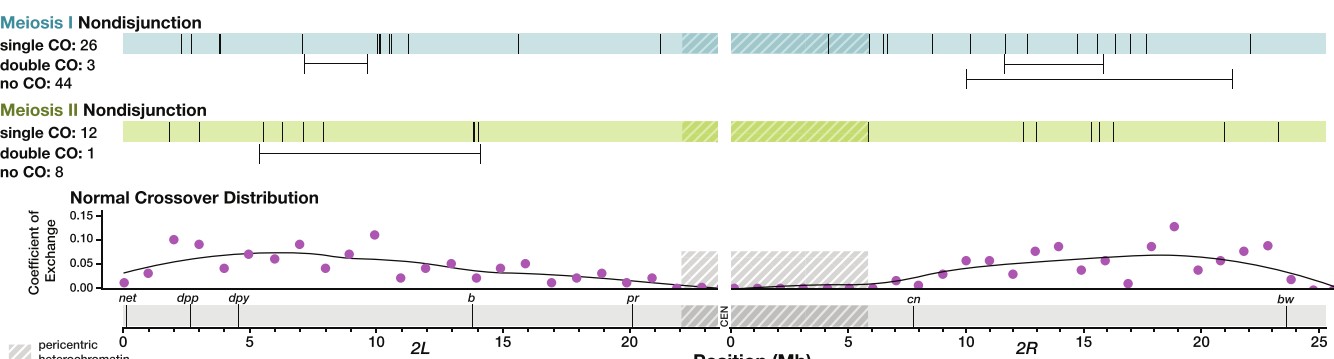

**Fig. 2.** Crossover distribution on chromosome 2 appears similar between normal and nondisjunctional meioses. Each line along the blue and green bars represents the location in megabases (Mb) of a single crossover (CO) along an arm of chromosome 2 in one NDJ male. Double crossovers are indicated below the bars, with horizontal lines depicting the distance between crossovers that occurred in the same individual arm of chromosome 2. Phenotypic genetic markers commonly used to assay recombination rate are displayed along a scale of physical distance in Mb. Shaded regions indicate pericentric heterochromatin as defined by H3K9 trimethylation in somatic cells (Stutzman et al. 2024). For comparison, coefficient of exchange in normal meioses is shown for 1 Mb intervals along chromosome 2 (pink), with black lines representing a best fit of the data. Data for normal crossover distribution, including best fit lines, was obtained from Miller et al. (2016).

occurred near this boundary. Notably, all of these events occurred within assembled DNA content, and not the highly repetitive satellite sequences surrounding the centromere. Assuming there was not a double crossover in the highly repetitive sequence (no crossovers have ever been detected in this region, with the possible exception of a single event identified in a heterochromatin mutant Miller et al. 2016; Hartmann et al. 2019; Pazhayam et al. 2025) it would be possible to detect events in the unassembled sequence from a shift in allele frequency between the end of the chromosome 2L assembly and the start of the chromosome 2R assembly. We did not identify any such cases in our dataset (Fig. 2 and Supplementary Table S3 in File S1).

Three MI NDJ males and one MII NDJ male had double crossovers on chromosome 2. All of these events had a distance between the left and right crossover of over 2 megabases (Mb), the smallest distance between double crossovers identified in Miller et al. (2016). Furthermore, all of these double crossovers occurred within the most highly recombining parts of the chromosome, with over a 10 centimorgan (cM) distance between the left and right crossovers for all four events on the standard genetic map. This suggests that these double crossovers are not abnormal in distance or location.

We also asked whether there was a relationship between number of crossovers on the left versus right arm of chromosome 2 in any single NDJ male (Supplementary Fig. S2 in File S1). For normal meioses, chromosomes with one or more crossovers on the left arm had a modestly lower chance of having a crossover on the right arm compared to chromosomes with zero crossovers on the left arm (52.7%, $n = 66$ vs. 66.7%, $n = 97$). A lower fraction of MI NDJ events had a crossover on the right arm regardless of the number of crossovers on the left arm: 44.4% of progeny with zero crossovers on the left arm of chromosome 2 had a crossover on the right arm ($n = 36$), and 38.5% of progeny with one or more crossovers on the left arm had a crossover on the right arm ($n = 13$). For MII NDJ, the trend was inverse of normal, with a higher fraction of males that had a crossover on the left arm having a crossover on the right arm (60.0%, $n = 4$) compared to males that did not have a crossover on the left arm (50.0%, $n = 10$). However, the sample size for MII NDJ was very limited and the number of crossovers that will be observed for MII NDJ is higher than that of MI NDJ or normal meioses due to MII NDJ progeny inheriting both of the chromatids of one homolog.

Given that the overwhelming majority of crossovers on chromosome 2 occurred within highly recombining parts of the genome, erroneous crossover placement may not be a major contributor to meiotic NDJ on chromosome 2. Additionally, the striking differences in recombination between MI and MII NDJ seen for the X chromosome are not apparent in NDJ of the $2^{nd}$ chromosome.

## Crossovers are distributed normally on the correctly segregating chromosomes in chromosome 2 meiotic NDJ

Nearly all previous studies of NDJ have assayed recombination exclusively on the nondisjoined chromosome. Whether recombination-related NDJ results from crossover patterning failures that are localized to a single chromosome, or whether it stems from a genome-wide disruption in crossover patterning remains unknown. Using whole-genome sequencing, we were able to measure recombination on the X and $3^{rd}$ chromosomes (Fig. 3). We did not analyze recombination on the small $4^{th}$ chromosome, which does not normally undergo crossing over and which was not isogenized in our parental genotypes. To maximize coverage of chromosome 2 we performed adaptive sampling in our sequencing, a technique enabling enrichment or depletion of specific DNA fragments by sequence content (Loose et al. 2016). Because we enriched for chromosome 2, most samples had considerably lower coverage of chromosomes X and 3. For each chromosome, we chose to analyze only the NDJ males for which there was high enough coverage to identify recombination events (or lack thereof) on that chromosome by eye ($n = 24$ for X, $n = 32$ for 3).

The X chromosome was enriched for crossovers along the medial portion of the chromosome arm, with a dearth of pericentric recombination in both the males that had MI or MII NDJ of the $2^{nd}$ chromosome (Fig. 3a). Indeed, the only observable recombination that occurred in the most proximal third of the chromosome was a single crossover between the commonly used phenotypic markers $f$ and $su(f)$, still well within the euchromatin, in a MI NDJ fly. One double crossover was observed in both MI and MII NDJ. Both events had over a 5 Mb distance between the left and right crossovers, which is well over the distance for the shortest double crossover tract length seen in Miller et al. (2016) of less than 2

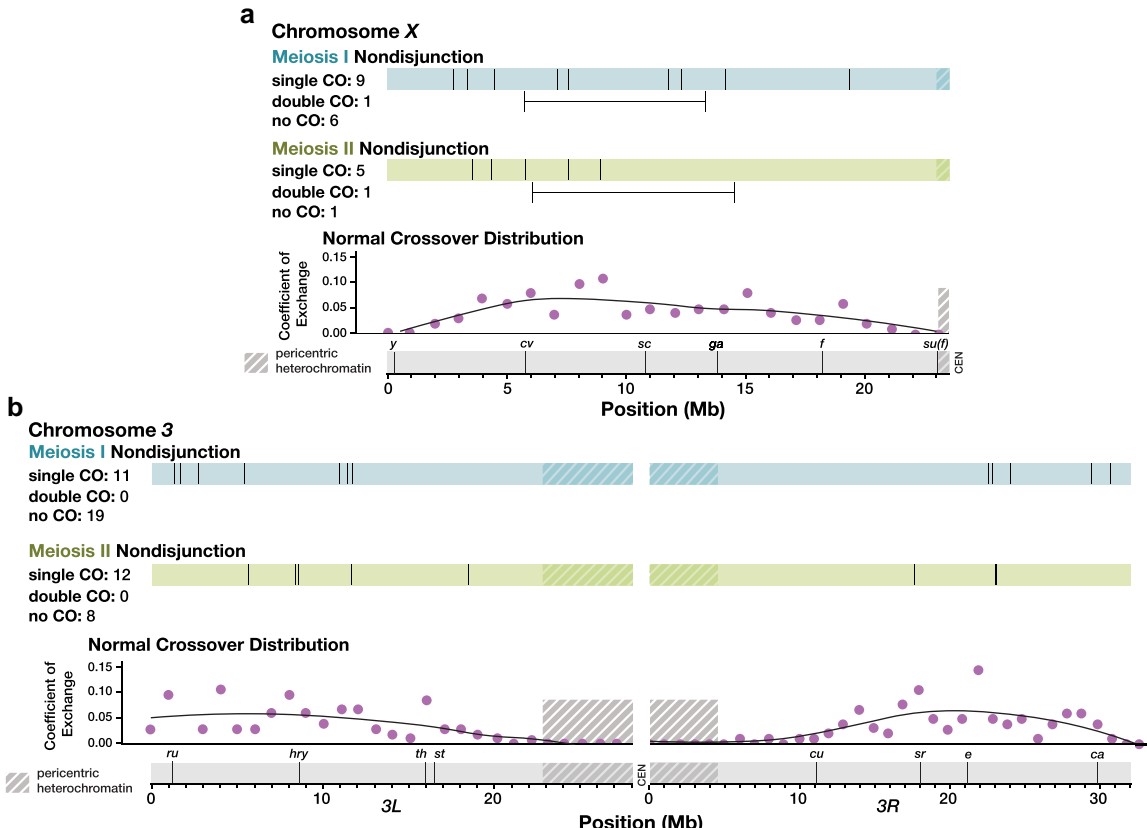

**Fig. 3.** Crossover distribution on chromosomes X and 3 appears similar between normal and chromosome 2 nondisjunctional meioses. Each line along the blue and green bars represents the location in Mb of a single crossover (CO) on one arm of chromosome X a) or chromosome 3 b), in chromosome 2 MI (blue) and MII (green) NDJ males. Double crossovers are indicated below the bars, with horizontal lines depicting the distance between crossovers that occurred in the same individual arm. Phenotypic markers frequently used to assay recombination rate are displayed along a scale of physical distance. Coefficient of exchange for normal meioses is shown for 1 Mb intervals along chromosomes X and 3 (pink), with black lines representing a best fit of the data. Shaded regions indicate pericentric heterochromatin as defined by H3K9 trimethylation in somatic tissue (Stutzman et al. 2024). Data for normal crossover distribution, including best fit lines, were obtained from Miller et al. (2016).

Mb. Additionally, both double crossovers occurred with a genetic map distance of over 25 cM between the left and right crossovers.

Crossover distribution on chromosome 3 was similar, albeit with some single crossovers in MI NDJ occurring in the distal portion of the chromosome arm, with three crossovers within the first 2 Mb of the assembly (Fig. 3b). Recombination in this area does not appear to be uncommon in normal meioses, as 18 crossovers occurred within this region in the Miller et al. (2016) dataset (with nine of these 18 occurring with another crossover elsewhere on the arm). Like the X chromosome, no pericentric crossovers on the $3^{rd}$ chromosome were observed in NDJ males. In sum, these results suggest that crossover distribution on the X chromosome and $3^{rd}$ chromosome are normal in both MI and MII NDJ of chromosome 2.

## MI NDJ events exhibit fewer crossovers on chromosome 2 as compared to normal meioses

Although placement of crossovers in chromosome 2 NDJ did not deviate strongly from normal crossover positioning, changes in crossover number may also contribute to NDJ. We consequently asked whether the fraction of MI and MII NDJ events with no observable crossover, a single crossover, or a double crossover on each arm of chromosome 2 differed from that of the normal meioses from Miller et al. (2016) (Fig. 4).

For this analysis, it must be noted that an unequal amount of information is inherited by the progeny of a normal meiosis, MI NDJ, and MII NDJ. Progeny of a normal meiosis will inherit a single chromatid out of the four. The probability of detecting a single crossover is 50%, as there is an equal likelihood of inheriting a recombinant or non-recombinant chromatid. MI NDJ progeny inherit one chromatid from each homologous chromosome. This results in a 50% chance of inheriting one recombinant and one non-recombinant chromatid, a 25% chance of inheriting reciprocal recombinant chromatids, and a 25% chance of inheriting two non-recombinant chromatids. In the latter two scenarios, it is impossible to detect a recombination event via changes in allele frequency, as both are fully heterozygous across the length of the chromosome. Therefore, the likelihood of detecting a single crossover in a MI NDJ event is also 50%. In contrast, because MII NDJ progeny inherit both sister chromatids from one chromosome, it is possible to detect all inter-homolog crossovers. We chose to handle this discrepancy by halving the fraction of MII NDJ single crossover events to equitably compare the three classes of events (Fig. 4, dotted line). We did not perform a similar adjustment for double crossovers, as they were rare overall.

A significantly lower fraction of MI NDJ chromosome 2 arms exhibited single crossovers relative to those of the normal meioses ($P = 0.009$, one-tailed two-proportion Z-test). The proportion of MI NDJ entire $2^{nd}$ chromosomes with zero detectable crossovers on either arm was over double that of normal meioses (13.4% vs. 40.8% of entire $2^{nd}$ chromosomes from normal and MI NDJ progeny, respectively), suggesting that the presence of at least one crossover per bivalent confers a better likelihood of correct

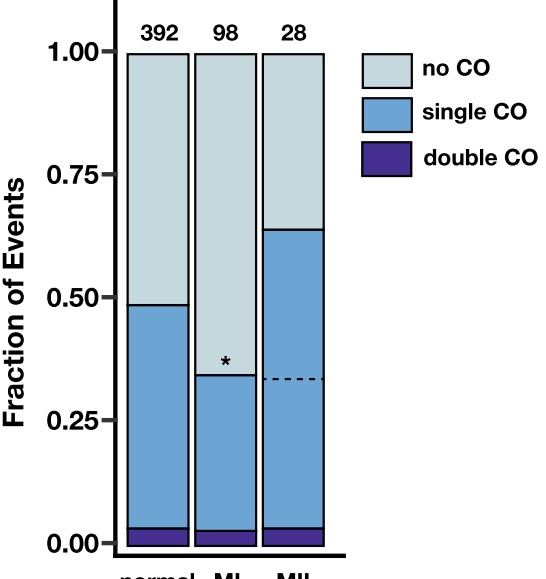

**Fig. 4.** Crossovers on chromosome 2 are reduced in chromosome 2 meiotic nondisjunction. The fraction of chromosome 2 arms that exhibited no observable crossovers (COs, light blue), one observable crossover (medium blue), and two observable crossovers (dark blue) is shown for normal meioses, MI nondisjunction of chromosome 2, and MII nondisjunction of chromosome 2. Numbers above bars indicate number of chromosome arms assayed per category. Because only half of single crossovers are observed for a normal meiosis (where there is only information from a single chromatid) and MI nondisjunction (where there is information from two chromatids from homologous chromosomes), but all single crossovers are observed for MII nondisjunction (where both chromatids from a single chromosome are inherited), a dotted line marking half of the MII single crossovers is shown for ease of comparison to the other categories. * denotes $P < 0.01$ by a one-tailed two-proportion Z-test.

homolog segregation. After adjustment, the fraction of MII NDJ arms with single crossovers was nearly identical to that of the MI NDJ arms, but was not significantly lower than that of the normal meioses, likely due to lower sample size ($P = 0.088$, one-tailed two-proportion Z-test). These data suggest that reduced recombination could be a contributing factor to meiotic NDJ of chromosome 2, similarly to the X chromosome, despite their other differences in recombination landscape in the context of NDJ. Unlike the X chromosome, which experienced reduced recombination in MI NDJ, but elevated recombination in MII NDJ, the fraction of chromosome 2 arms that had single crossovers appears to be similar between MI and MII NDJ. This observation is consistent with the similarity in recombination landscape between MI and MII NDJ on chromosome 2.

Middlebrooks et al. (2014) found a global reduction in recombination rate in MI nondisjunction of human chromosome 21 when there were one or fewer crossovers on the nondisjoined chromosome. Therefore, we asked whether number of crossovers on chromosome 2 was correlated with mean number of crossovers on chromosomes X and 3 (Supplementary Fig. S3 in File S1). We only considered MI events for this analysis due to the difference in probability of observing a crossover on chromosome 2 in MI versus MII NDJ and low number of MII samples overall. The mean number of crossovers on other chromosomes was the same when either zero or one crossovers were observed on chromosome 2 ($P = 0.391$, two-tailed Mann-Whitney test) (Supplementary Fig. S2 in File S1). The increase in recombination

rate on other chromosomes was not significant when two crossovers were observed on chromosome 2 compared to when one ($P = 0.072$, one-tailed Mann-Whitney test) or zero ($P = 0.071$, one-tailed Mann-Whitney test). These data could indicate that lower recombination on chromosome 2 in MI NDJ is associated with globally reduced recombination, but are not conclusive.

## Discussion

Our findings suggest that the differences between NDJ on chromosome 2 and NDJ on the previously studied X chromosome are three-fold: 1) crossovers, while modestly reduced in chromosome 2 NDJ, are not drastically altered in their positioning, 2) MII events, while still less common than MI events, comprise a greater proportion of chromosome 2 NDJ than they do X chromosome NDJ, and 3) chromosome 2 NDJ is not associated with stark differences in crossover patterning between MI and MII NDJ. This raises the question: Why is meiotic NDJ of chromosome 2 so different from that of the X chromosome?

The X chromosome and human acrocentric chromosomes follow similar trends, with high incidence of MI NDJ, low recombination rate in MI NDJ, and elevated pericentric recombination in MII NDJ (Lamb et al. 1996, 1997; Robinson et al. 1998). This is in contrast to chromosome 2 and the few human metacentric and sub-metacentric chromosomes that have been studied (Hassold et al. 1995; Bugge et al. 1998). This suggests that the factors contributing to NDJ of acrocentric or telocentric chromosomes differ from those of metacentric chromosomes. We suggest that three key recombination patterns may confer greater risk of NDJ in either a telocentric or metacentric chromosome: the absence of crossovers, distal crossovers, and pericentric crossovers.

Based on data from viable progeny, the X chromosome fails to receive a crossover roughly as often as each arm of chromosome 2 (7–12% of meioses vs. 8–16% of meioses, Baker and Carpenter 1972; Parry 1973; Page et al. 2007; Miller et al. 2016). Assuming a crossover on at least one arm is sufficient for proper disjunction, this would suggest that chromosome 2 has a lower risk of NDJ due to lack of a chiasma compared to the X chromosome (Fig. 5a). Similarly, a metacentric chromosome would be at greater risk of NDJ due to a pericentric crossover, as there are now two arms at risk of having a pericentric crossover. Finally, a single distal crossover may be less effective in ensuring segregation of an acro/telocentric chromosome, as a distal crossover on one arm of a metacentric may be rescued by a medial crossover on the other arm.

If these hypotheses are true, the proportion of X chromosome NDJ with no observable crossover or a single distal crossover should be greater than that of the $2^{nd}$ chromosome, and the proportion of X chromosome NDJ with pericentric crossovers should be reduced compared to that of the chromosome 2 NDJ. To test this, we took the NDJ progeny from our dataset and that of Koehler et al. (1996a) and categorized them based on the most probable reason for their NDJ. Firstly, any NDJ progeny with no observable recombination on the nondisjoined chromosome were categorized as having experienced NDJ due to lack of a chiasma. Secondly, NDJ progeny with a crossover in the distal 24% of the chromosome (the percentage of the X chromosome made up by the distalmost interval used to measure recombination in Koehler et al. 1996a), with no detectable medial or proximal crossovers, were categorized as having experienced NDJ due to a distal crossover. Thirdly, we categorized any NDJ progeny with a crossover within the pericentric heterochromatin (marked by su(f) to y+ in Koehler et al. 1996a) or 15% of the adjacent euchromatin

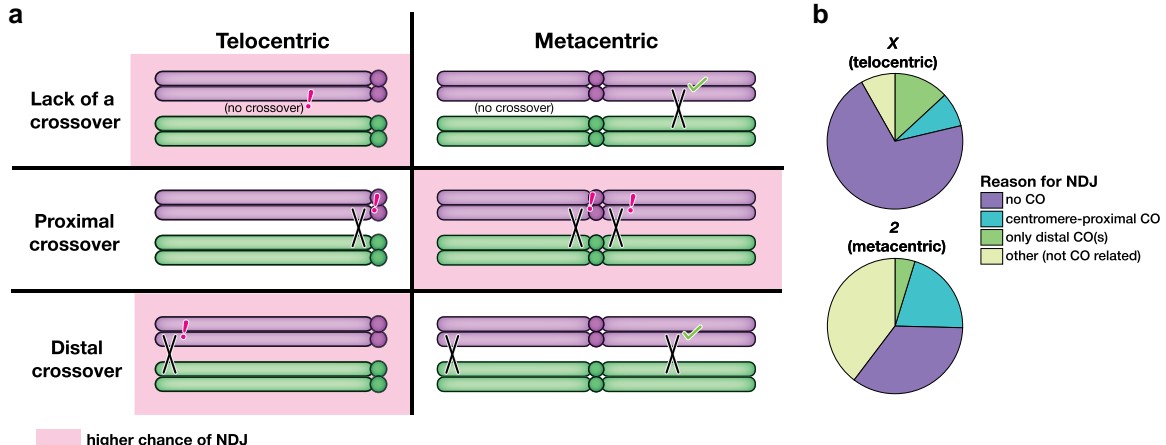

**Fig. 5.** Differential causes of nondisjunction in telocentric versus metacentric chromosomes. a) Crossover patterning errors that may lead to NDJ. Pink shading indicates that a certain pattern of recombination may confer a higher chance of NDJ for that chromosome shape (telo- vs. metacentric). The absence of crossovers or a sole distal crossover on a chromosome arm may be more deleterious for a telocentric chromosome than a metacentric, where it is likely thatt there is a crossover on the other arm. In comparison, metacentric chromosomes may be at higher risk of NDJ due to pericentric crossovers as there are two pericentric regions as opposed to one on a telocentric chromosome. b) A breakdown of NDJ events for the *X* chromosome and the 2[nd] chromosome based on the most likely cause of NDJ.

(the percentage of the *X* chromosome made up by *car* to *su(f)*) as having experienced NDJ due to a pericentric crossover. Finally, all progeny that did not fall into the three aforementioned categories were classified as having experienced NDJ due to factors unrelated to recombination.

The data binned accordingly were consistent with our predictions (Fig. 5b). The percentage of NDJ progeny with no observable crossover and the percentage of NDJ progeny with a distal crossover were elevated for *X* chromosome NDJ relative to chromosome 2 NDJ. Additionally, the percentage of NDJ progeny with pericentric crossovers was reduced for *X* chromosome NDJ. The fraction of events that appeared to be unrelated to recombination was also greatly elevated for chromosome 2 NDJ.

A caveat of our experimental design is that it is not possible to recover progeny from normal disjunction of chromosome 2. By consequence, there is no way to determine if the rate of chromosome 2 meiotic NDJ is similar to that of *X* chromosome NDJ. While it is apparent that a greater proportion of chromosome 2 NDJ is MII in nature compared to *X* chromosome NDJ (Fig. 1b), it is unclear if this is due to MI events being reduced, with the rate of MII NDJ being similar between the two chromosomes, or if the rate of MII NDJ is elevated for chromosome 2 relative to the *X*. Similarly, while a greater proportion of chromosome 2 NDJ appears to be unassociated with recombination (Fig. 5b), we cannot say whether these NDJ events that are unrelated to recombination occur more frequently in chromosome 2 NDJ or if recombination-related NDJ is simply less frequent.

Middlebrooks et al. (2014) found that when human MI NDJ chromosomes had zero or one crossovers there were fewer crossovers on other chromosomes. However, crossovers were not reduced genome-wide in the case of MII errors. We did not observe a significant elevation in mean recombination rate on other chromosomes for MI NDJ progeny when there were two crossovers on chromosome 2 versus when there were zero or one (Supplementary Fig. S2 in File S1). However, it is possible that more NDJ events with a global reduction in recombination rate exist in *Drosophila*, but our experimental design precludes us from observing them when this reduction results in NDJ of other chromosomes in conjunction with chromosome 2. 26 of 103 of the male NDJ progeny obtained in our experiment were sterile. A possible explanation for this

sterility is that these are *X0* males resulting from simultaneous NDJ of chromosomes *X* and 2.

Unlike the metacentric human chromosomes for which recombination in NDJ has been studied, *D. melanogaster* chromosomes 2 and 3 are very similar in size, amount of heterochromatin, and shape (Adams et al. 2000). Future work exploring NDJ of chromosome 3 would help to parse whether these factors are major contributors to the discrepancy in recombination landscape in NDJ of the metacentric human chromosomes.

## Data availability

Sequencing data from this study can be found under the National Center for Biotechnology Information (NCBI) BioProject number PRJNA1398379.

Supplemental material available at GENETICS online.

## Acknowledgments

We thank Susan McMahan and Evan Dewey for their assistance in collecting flies for the genetic crosses required to obtain NDJ progeny.

## Funding

This work was supported by a grant from the National Institute of General Medical Sciences to J.S. under award 1R35GM118127. C.T. was supported in part by grants from the National Institute on Aging under award 1F31AG074637 and the National Institute of General Medical Sciences under award 5T32GM007092-45.

## Conflicts of interest

The authors have no conflict of interest to declare.

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
