## [Peer Review File · Genetics]

Chromosome-specific differences in the recombination landscape of spontaneous meiotic nondisjunction

Carolyn Turcotte and Jeff Sekelsky

NOTE: The reviews and decision letters are unedited and appear as submitted by the reviewers.

In extremely rare instances and as determined by a Senior Editor or the EIC, portions of a review may be redacted. If a review is signed, the reviewer has agreed to no longer remain anonymous.

The review history appears in chronological order.

Review Timeline:

Submission Date:	2026-01-06
Editorial Decision:	2026-02-15
Revision Received:	2026-03-04
Accepted:	2026-03-09

February 15, 2026

RE: GENETICS-2026-308950

Dear Dr. Sekelsky:

I am pleased to accept your manuscript titled "Chromosome-specific differences in the recombination landscape of spontaneous meiotic nondisjunction" for publication in GENETICS, pending minor revision.

Please submit your revision along with a brief description of how you modified the manuscript in response to the reviewers' concerns and suggestions (which can be viewed at the bottom of this email. Both of the reviewers found the study to be a valuable contribution to our understanding of meiotic segregation (as did I), and their comments focus mainly on presentation/explanation of data and clarifications about models and predictions. A suitably revised manuscript will be acceptable for publication; I don't expect to send it out for review.

When revising the ms., please make an effort to shorten it, because that almost always improves a manuscript. We urge authors to heed the advice of Strunk and White: "omit needless words"¹. Follow this link to submit the revised manuscript: Link Not Available

Thank you for submitting this story to Genetics.

Sincerely,

Jack Bateman
Associate Editor
GENETICS

Approved by:
Amy MacQueen
Senior Editor
GENETICS

Reviewer comments:
Reviewer #1 :

The authors used a compound autosome to recover progeny that resulted from the nondisjunction of the second chromosome in *D. melanogaster*. They then used whole genome sequencing to identify and position the recombination events that occurred during those nondisjunctive female meioses. This type of experiment had previously been performed for the X chromosome, so the authors compared their results to those previous studies, and found significant differences between these two chromosomes in the abundance of meiosis II events and how crossovers were distributed across the chromosome. One caveat is that while studying X nondisjunction allows for the recovery of both normal and nondisjunctive progeny, the use of the compound 2 chromosome means that all normal progeny will die (from either monosomy or trisomy for a major autosome), which complicates the analysis by not allowing a denominator for making direct rate comparisons between the chromosomes.

I thought the paper was interesting to read, and the results extend our knowledge of the events that lead to nondisjunctive meiosis, so would be of interest to readers of Genetics. I really couldn't find much to criticize about the manuscript; really the only thing I was not clear on was in the position maps in figures 2 and 3. The legend says that "coefficient of exchange in normal meioses is shown for 1 Mb intervals", and I am assuming that those pink dots above the position figure are those coefficients. But there is also a black throughline on those figures, and I'm not sure what that is supposed to represent or how it was calculated.

As I write this, I guess I also am not sure how their CO position data (the tick marks above the position map) integrates into the distribution of CO events -- I can give the authors the benefit of the doubt that the distribution of COs is more normal in 2 NDJ than for X NDJ, its just from reading their paper I'm not sure how to go from their data to that conclusion. This lack of

understanding could very well be a deficiency in the reader, and if I just put in more time to study the manuscript I might be able to figure it out, but some more guidance on how to get there would be welcome.

Reviewer #2 :

This is a very nice contribution to our understanding of the way in which meiotic recombination influences chromosomal segregation. I only have a few minor comments.

1. I will admit, it is fairly challenging thinking of all the different ways in which different events (NCO M1 NDJ, CO M2 NDJ, mitotic NDJ) can be detected with the methods. After closely reading and diagraming things myself, it is clear that their approaches are correct. For example, the method of reducing the CO count in the M2 NDJ seems correct. That being said, it might be worthwhile for the reader to have a supplemental figure that explains the reasoning in visual manner.

2. It isn't clear why they don't do any statistics comparing placement of COs across the conditions. They should be able to compare placement of the single COs statistically between M1 NDJ events, M2 NDJ events and those from Miller et al. 2016 (are those placements available). One could either do a t-test for positions or some sort of permutation. Their interpretation seems correct - that the distributions of CO positions are the same. But it would be worth testing this formally.

3. I am a little confused about the prediction that being metacentric leads to INCREASED susceptibility to NDJ being caused by cen co events. Perhaps part of the issue is from figure 5. For the example of proximal crossing over, there are TWO proximal events diagrammed. But they don't seem to be saying that the occurrence of two is specifically the problem. Rather, they seem to imply the problem arises when there is AT LEAST ONE proximal event. They don't frame this as an "at least one" probability. They should say something explicitly such that the probability of AT least one for one arm (the X) is $(1 - \text{probability of distal})$ and the probability of at least one for the two arm 2nd is $(1 - (\text{prob of distal arm 1}) \text{ AND } (\text{prob of distal arm 2})) = (1 - p^2)$ vs $(1 - p)$. This may clarify things for the reader. They should also explain how AT least one event might be problematic mechanistically rather than having BOTH proximal in order for NDJ to increase.

4. Related to this, it is a little hard to reconcile their reasoning on these matters in the discussion with respect to the pie charts in 5B. If the proportion of NDJ events in a metacentric is lower (than for the X) for the NO CO class, won't by definition the proportion of centromere-proximal causes be increased, all else being equal? Maybe I am wrong, because the not CO related class seems to increase dramatically. But the only distals also goes down. In fact, you can kind of see that for both, 3/4 of the pie is either no-CO or other and 1/4 of the pie is centromere-proximal or only distal. Just focusing within this 1/3 of the two pies, if the only distals class goes down, by necessity, the centromere proximal CO cause will go up. Overall, there is a non-independence among these estimates since the proportion has to add to 1.0 across the different reasons. Some consideration of this issue is warranted.

Reviewer #1:

The authors used a compound autosome to recover progeny that resulted from the nondisjunction of the second chromosome in *D. melanogaster*. They then used whole genome sequencing to identify and position the recombination events that occurred during those nondisjunctional female meioses. This type of experiment had previously been performed for the X chromosome, so the authors compared their results to those previous studies, and found significant differences between these two chromosomes in the abundance of meiosis II events and how crossovers were distributed across the chromosome. One caveat is that while studying X nondisjunction allows for the recovery of both normal and nondisjunctional progeny, the use of the compound 2 chromosome means that all normal progeny will die (from either monosomy or trisomy for a major autosome), which complicates the analysis by not allowing a denominator for making direct rate comparisons between the chromosomes.

I thought the paper was interesting to read, and the results extend our knowledge of the events that lead to nondisjunctional meiosis, so would be of interest to readers of Genetics. I really couldn't find much to criticize about the manuscript; really the only thing I was not clear on was in the position maps in figures 2 and 3. The legend says that "coefficient of exchange in normal meioses is shown for 1 Mb intervals", and I am assuming that those pink dots above the position figure are those coefficients. But there is also a black throughline on those figures, and I'm not sure what that is supposed to represent or how it was calculated.

We have adjusted the legends of Figures 2 and 3 as follows: "Coefficient of exchange in normal meioses is shown for 1 Mb intervals (pink), with black lines representing a best fit of the data... Data for normal crossover distribution, including best fit lines, were obtained from Miller et al. (2016)"

Unfortunately, Miller et al. do not describe the method used to produce this best fit, so we are unable to provide further detail. However, the best fit line from this data tracks well with previous measures of recombination in wild-type *Drosophila* that were obtained from phenotypic mapping, suggesting that it is appropriate to be used as a representation of normal crossover distribution.

As I write this, I guess I also am not sure how their CO position data (the tick marks above the position map) integrates into the distribution of CO events -- I can give the authors the benefit of the doubt that the distribution of COs is more normal in 2 NDJ than for X NDJ, its just from reading their paper I'm not sure how to go from their data to that conclusion. This lack of understanding could very well be a deficiency in the reader, and if I just put in more time to study the manuscript I might be able to figure it out, but some more guidance on how to get there would be welcome.

Due to the low number of data points for chromosome 2 nondisjunction, it is difficult to directly compare the distribution of crossovers in chromosome 2 NDJ to normal crossover distribution. However, we do not observe drastic shifts in the locations of crossovers in chromosome 2 NDJ, unlike what was observed for X chromosome NDJ in Koehler 1996. To clarify this point, we have edited the text as follows: "Given that the overwhelming majority of crossovers on chromosome 2 occurred within highly recombining parts of the genome, crossover placement may not be a major contributor to meiotic NDJ on chromosome 2." (p. 4, line 51).

Reviewer #2 :

This is a very nice contribution to our understanding of the way in which meiotic recombination influences chromosomal segregation. I only have a few minor comments.

1. I will admit, it is fairly challenging thinking of all the different ways in which different events (NCO M1 NDJ, CO M2 NDJ, mitotic NDJ) can be detected with the methods. After closely reading and diagraming things myself, it is clear that their approaches are correct. For example, the method of reducing the CO count in the M2 NDJ seems correct. That being said, it might be worthwhile for the reader to have a supplemental figure that explains the reasoning in visual manner.

We agree that it is difficult to visualize the possible ways that NDJ may manifest in progeny, so we added a new Supplemental Figure 1, which provides visual representations of how MI and MII NDJ would manifest in progeny after a single CO on one chromosome arm, as well as two examples of monosomy rescue from mitotic NDJ, demonstrating, for example, how COs in MI NDJ may not be detectable in progeny and how monosomy rescue can appear visually indistinguishable from MII NDJ.

2. It isn't clear why they don't do any statistics comparing placement of COs across the conditions. They should be able to compare placement of the single COs statistically between M1 NDJ events, M2 NDJ events and those from Miller et al. 2016 (are those placements available). One could either do a t-test for positions or some sort of permutation. Their interpretation seems correct - that the distributions of CO positions are the same. But it would be worth testing this formally.

Unfortunately, the limited number of data points makes it challenging to formally compare the distribution of crossovers for chromosome 2 NDJ to the Miller 2016 dataset. Making meaningful comparisons of crossover distributions obtained using phenotypic markers, for example, requires several thousand flies, which is not feasible in the context of NDJ in wild-type flies (e.g., X chromosome meiotic NDJ occurs in <0.01% of meioses that form viable progeny). We consulted with someone in the Department of Statistics and Operations Research, and they did not have a solution for us.

3. I am a little confused about the prediction that being metacentric leads to INCREASED susceptibility to NDJ being caused by cen co events. Perhaps part of the issue is from figure 5. For the example of proximal crossing over, there are TWO proximal events diagrammed. But they don't seem to be saying that the occurrence of two is specifically the problem. Rather, they seem to imply the problem arises when there is AT LEAST ONE proximal event. They don't frame this as an "at least one" probability. They should say something explicitly such that the probability of AT least one for one arm (the X) is $(1 - \text{probability of distal})$ and the probability of at least one for the two arm 2nd is $(1 - (\text{prob of distal arm 1}) \text{ AND } (\text{prob of distal arm 2})) = (1 - p^2)$ vs $(1 - p)$. This may clarify things for the reader. They should also explain how AT least one event might be problematic mechanistically rather than having BOTH proximal in order for NDJ to increase.

We regret the confusion the figure caused. As suggested, we've have rearranged the text in the paragraph referencing Figure 5A and edited the portion about pericentric COs as follows: "Similarly, a metacentric chromosome would be at greater risk of NDJ due to a pericentric crossover, as there are now two arms at risk of having a pericentric crossover." (p. 6, line 73)

4. Related to this, it is a little hard to reconcile their reasoning on these matters in the discussion with respect to the pie charts in 5B. If the proportion of NDJ events in a metacentric is lower (than for the X) for the NO CO class, won't by definition the proportion of centromere-proximal causes be increased, all else being equal? Maybe I am wrong, because the not CO related class seems to increase dramatically. But the only distals also goes down. In fact, you can kind of see that for both, 3/4 of the pie is either no-CO or other and 1/4 of the pie is centromere-proximal or only distal. Just focusing within this 1/3 of the two pies, if the only distals class goes down, by necessity, the centromere proximal CO cause will go up. Overall, there is a non-independence among these estimates since the proportion has to add to 1.0 across the different reasons. Some consideration of this issue is warranted.

The reviewer is correct that the different categories are expressed as fractions of the total for the two datasets (published X ndj and our 2 ndj), and therefore if one category accounts for a greater proportion of chromosome 2 ndj, then one or more other categories must necessarily go down. We feel our discussion of this point is clear, since we use the word "proportion" throughout. We also point out (p. 7, line 23) that use of a compound chromosome means that only ndj progeny survive, so we can't determine the absolute frequency of any type of event, just the proportion of each category among the ndj survivors.

March 9, 2026

RE: GENETICS-2026-308950R1

Dr. Jeff Sekelsky
The University of North Carolina at Chapel Hill
Department of Biology
CB #3280, 303 Fordham Hall
Chapel Hill, North Carolina 27599-3280

Dear Dr. Sekelsky:

Congratulations, your manuscript titled "Chromosome-specific differences in the recombination landscape of spontaneous meiotic nondisjunction" is accepted for publication in GENETICS! Many thanks for submitting your research to the journal.

To Proceed to Publication:

1. Format your article according to GENETICS style: <https://academic.oup.com/genetics/pages/author-guidelines>
2. Ensure that you comply with data and community resource citation guidelines: <https://academic.oup.com/genetics/pages/author-guidelines#section-5-9-2>
3. Upload your final files at <https://genetics.msubmit.net>
4. Add oupsupport@scipris.com and genetics.oup@novatechset.com (or the domains @scipris.com and @novatechset.com) to your email program's "safe senders" list. You will be contacted by both at various points during the production process.

Notes:

- Your currently-accepted manuscript (unedited, as submitted, reviewed, and accepted) will be published at GENETICS and deposited into PubMed as an Advance Access article. Notify sourcefiles@thegsajournals.org before signing your license if you do not wish to publish your article via Advance Access.
- We invite you to submit an original color figure related to your paper for consideration as cover art. Please email your submission to the editorial office or upload it with your final files. You can submit a small-sized image for evaluation, and if selected, the final image must be a TIFF file 2513px wide by 3263px high (8.375 by 10.875 inches; resolution of 600ppi). Please avoid graphs and small type.
- After files are sent to Oxford University Press we use SciPris to manage article licensing and payment. If you do not have a SciPris account, you will receive an email from no-reply@scipris.com to sign up to use Oxford University Press' author portal. After logging in, follow the online instructions to sign your license and arrange any payment due.

If you have any questions or encounter any problems while uploading your accepted manuscript files, please email the editorial office at sourcefiles@thegsajournals.org.

Sincerely,

Jack Bateman
Associate Editor
GENETICS

Approved by:
Amy MacQueen
Senior Editor
GENETICS